# Does *Opuntia ficus-indica* Juice Supplementation Improve Biochemical and Cardiovascular Response to a 6-Minute Walk Test in Type 2 Diabetic Patients?

**DOI:** 10.3390/medicina58111561

**Published:** 2022-10-30

**Authors:** Salma Abedelmalek, Khouloud Aloui, Nesrine Boussetta, Bayan Alahmadi, Mohamed Zouch, Hamdi Chtourou, Nizar Souissi

**Affiliations:** 1Laboratory of Physiology and Functional Exploration, Faculty of Medicine, Sousse 4002, Tunisia; 2Department of Sport Science and Physical Activity, College of Education, University of Ha’il, Hail 55255, Saudi Arabia; 3Physical Activity, Sport, and Health, UR18JS01, National Observatory of Sport, Tunis 1003, Tunisia; 4High Institute of Sport and Physical Education of Sfax, University of Sfax, Sfax 3000, Tunisia

**Keywords:** OFIJ supplementation, 6MWT, diabetes, muscle damage, biochemical parameters

## Abstract

*Background and objectives:* The purpose of this study was to evaluate the effect of *Opuntia ficus-indica* juice (OFIJ) on performance and biochemical and physiological responses to a 6 min walking test (6MWT) in diabetic patients. *Materials and Methods:* Twenty diabetic patients performed a 6MWT at 07:00 h. During each test session, they were asked to drink 70 mL/day of natural OFIJ or placebo (PLA) for 4 days. *Results:* the results showed that cardiovascular parameters increased significantly after the 6MWT under both conditions. While, cortisol, HbA1c, cholesterol total (CT), triglycerides (TG), as well as low-density lipoprotein (LDL) were not modified between without and with supplementation. Likewise, no significant variation in performance was observed for PLA and OFIJ (*p >* 0.05). The cardiovascular parameters (heart rate max (HRmax), diastolic blood pressure (DBP), and systolic blood pressure (SBP)), lipid profile (CT, TG, LDL, and high-density lipoprotein HDL), hormonal parameters (insulin and glucagon), HbA1c and lactate ([La]) did not present any significant modification either between PLA or OFIJ (*p* > 0.05). Muscle-damage markers (creatine kinase (CK) and lactate dehydrogenase (LDH)], cortisol, and liver parameters (i.e., oxidative stress marker, γGT, and total bilirubin) as well as glucose (GLC) were affected by supplementation (*p <* 0.05) before and after the 6MWT, but this change was significant only for OFIJ (*p <* 0.05). *Conclusion*: OFIJ had an antioxidant capacity, improved performance of the 6MWT, and reduced muscle-damage markers and glucose level in type 2 diabetic patients.

## 1. Introduction

According to the World Health Organization, diabetes is a major public health problem. The global prevalence of type 2 diabetes in adults has increased from ~150 million affected people in 2000 to >450 million in 2019 and is projected to rise further to ~700 million by 2045 [1]. To prevent complications of diabetes and for early management, a better knowledge of the lifestyle of diabetics and their eating and physical-activity habits must be developed through a monitoring system. Furthermore, diabetes is on the rise around the world, and it is linked to slew of metabolic and clinical disorders that result in a higher risk for cardiovascular or metabolic diseases. In particular, the development of insulin resistance and type 2 diabetes is closely related to obesity and inactivity. As a result, sport is considered in all treatment guidelines as a basic requirement for successful diabetes therapy [2,3]. Thus, the benefits of exercise in terms of improving glycaemic control, reducing cardiovascular risk factors, and preventing complications associated with diabetes are clearly documented for people with type 2 diabetes. In addition, moderate physical activity combined with a better dietary balance can delay the onset of type 2 diabetes [4]. It is known that physical exercise increases the production of reactive oxygen species (ROS) by cells, which can lead to cellular damage [5,6,7].

However, it is well established that increased consumption of flavonoids in most fruits and vegetables is beneficial to one’s health [8]. The Opuntia supplement (OF) has been administered in traditional medicines [9]. Additionally, OF is considered advantageously due to its high fiber content, particularly pectin, as well as its high mineral and phytochemical content. A recent meta-analysis study that used data from 11 studies concluded that Prickly Pear Fruit and OF consumption were associated with significant reductions in TC and LDL-C levels [10].

*Opuntia ficus-indica* (OFIJ) has beneficial effects on metabolism, glucose homeostasis, oxidative stress, total cholesterol (TC), low-density lipoprotein cholesterol (LDL), and high-density lipoprotein cholesterol (HDL) [11,12]. Over the last decade, several studies have determined the effectiveness of OFIJ in terms of nutrition and health [12,13]. Indeed, this fruit is used as potent therapy in a number of pathologies (i.e., diabetes, hypercholesterolemia, obesity, cancer, and benign prostatic hypertrophy) [14]. OFIJ appears to be an excellent candidate for nutritional recommendations and therapeutic indications, based on its natural content of polyphenols, vitamins, and other specialized components. Clinical research on *Opuntia streptacantha* has confirmed that it has anti-hyperglycaemic properties, whereas clinical data on OFIJ are limited [12,15]. Given the fact that exercise have been proven to be a viable approach in preventing or lessening the risk of type 2 diabetes, OFIJ may represent a good therapy to limit diabetes risk. To the best of our knowledge, no study has examined the effect of OFIJ supplementation on physical and physiological parameters in patients with diabetes type 2.

Therefore, the purpose of this study was to examine the effects OFIJ on biochemical and physiological responses to a physical task. We hypothesized that OFIJ supplementation may improve distance during a 6 min walk test and result in better biochemical and physiological responses in type 2 diabetic patients.

## 2. Material and Methods

### 2.1. Participants

Twenty patients with type 2 diabetes (age (years): 55.4 ± 1.2; height (m): 1.7 ± 0.01; body composition (kg); 68.3 ± 1.7)) volunteered to participate in this study after the exclusion of nine participants because they did not meet the inclusion criteria. After receiving a detailed explanation of the protocol, they gave written consent to participate in the study. The participants were also selected based on their chronotype and their answers to the self-assessment questionnaire of Horne and Ostberg [16]. They had an intermediate chronotype (i.e., sleep duration between 23:00 h ± 1:00 h and 07:00 h ± 1:00 h). The exclusion criteria included people with known health problems such as hypertension or heart, kidney, or liver problems. In addition, people who regularly took medications or hypoglycemic drugs, including dietary supplements or vitamins, were not eligible for the study. The study protocol was in line with the Helsinki Declaration for human experimentation and was approved by the University Ethics Committee of Tunisia.

### 2.2. Experimental Design

The experimental procedure is illustrated in Figure 1. During the 4 days of treatment, supplementation of 70 mL/daily of OFIJ was utilized. One week before the start, the patients were familiarized with the experimental material and protocol to overcome the effects of learning that could occur with the repetition of the test sessions [17]. To limit the influence of exogenous factors on blood parameters or performance, indications were given to each subject before the experiment. Patients were asked to use the same clothing and the same sports shoes at each session and to not use antioxidants or stimulants or perform strenuous activity during the 24 h before the experiment [18]. The patients were to be in the field in the morning (07:00 h) in a fasted state and only one glass of water was allowed [19] to overcome the postprandial effects characterized, in particular, by an increase of the central temperature.

The experimental procedure consisted of performing two 6 min walk tests (6MWT) before and after OFIJ supplementation or a placebo solution (PLA). The rating of perceived exertion (RPE) was recorded following the 6MWT. All test sessions were performed in the morning (07:00 h–09:00 h). Patients are invited to drink 70 mL/day of OFIJ for 4 days [20] or the same procedure with PLA.

Measurement of cardiovascular (i.e., HR, SBP, and DPB) parameters and blood samples were taken at rest and at the end of the 6MWT. The cardiovascular parameters of the patients were recorded using an electronic wrist blood-pressure monitor (Microlife, W90, Paris, France) [21].

### 2.3. Antiradical Capacity by DPPH

The 2,2-diphenyl-1-picryl-hydrazyl-hydrate (DPPH) test was used to predict antioxidant activities by a mechanism in which antioxidants act to inhibit lipid oxidation. This method was used as it is simple and sensible. DPPH assays were based on the rule described by Brand-Williams et al. [22], but the analytic protocols differed in some parameters (i.e., absorbance, reaction time, and the reference solution). DPPH, a purple-colored free radical formed the yellow-colored diphenyl-picryl-hydrazine. The method used in our study on OFIJ was that described by Tuberoso et al. [23]. An ethanolic solution (2.5 mg/100 mL) of the stable DPP0048 radical was prepared and applied for the assay. OFIJ was dissolved in 10 mL of ethanol, and placed into test tubes. Then, 250 µL of the ethanolic solution was added to samples. The test tubes were incubated for 30 min, and the optical density was read at 520 nm against the ethanol in a UV spectrometer (Genesys-10, Madison, WI, USA). The radical scavenging activity was estimated utilizing the following equation:% scavenging activity = ((A_control_ − A_sample extract_)/A_control_) × 100

In this study, the Student’s *t* test for independent samples showed a significant dose-dependent decrease (*t* = −8.95; *p <* 0.05) for OFIJ extracts compared with ethanol solution. All the extracts inhibited DPPH with a minimum of 10.3% during 30 min of incubation. Thus, OFIJ introduced an anti-radical activity that was significantly higher than the ethanol solution.

### 2.4. Sample and Treatments

*Opuntia ficus-indica* fruit with purple color was selected from a local market in Tunisia. Only fruit without external injuries was chosen. It was then washed and peeled manually. To extract the juice, the pulp was pressed using an industrial mixer (Kenwood, 700W, Shanghai, China) and was then drawn through a strainer to the remove seeds and to extract the maximum amount of juice. No additional chemicals were added to the juice. The placebo was a fruit concentrate of isoenergetic origin synthetically designed to have a similar consistency and color, but without the phytochemical content of OFIJ. Meanwhile, a 70 mL dose of OFIJ was selected, as previous studies had reported benefits using a similar concentration of antioxidants [20].

### 2.5. 6 Min Walk Test (6MWT)

The 6MWT was selected in the present study as it is a submaximal aerobic exercise that can be performed by patients with type 2 diabetes. Participants were instructed to walk as far as possible within 6 min, back and forth along a 30 m track. During the test, no hopping, skipping, sprinting, or jumping were permitted. The participants were only given standardized encouragement (e.g., ‘go ahead’) and an announcement of the remaining time.

#### Rating of Perceived Exertion (RPE)

The RPE scale allows participants to rate their level of exertion for a given physical task. At the end of the 6MWT, the RPE was recorded using the Borg Rating of Perceived Exertion Scale. The level of exertion ranges from 6 to 20.

### 2.6. Blood Samples and Analyses

A 5 mL sample of venous blood was taken from venous catheters at rest and at the end of the exercise. Blood was then centrifuged (for 20 min at 3000 rpm at room temperature) to separate the erythrocytes from the plasma and was frozen and stored at −80 °C until analysis.

Low-density lipoprotein (LDL) and high-density lipoprotein (HDL) were measured using a colorimetric enzymatic assay: homogeneous phase direct test (liquicolor) (Kit, HUMAN, Ref: 10084: Gesellschaft für Biochemica und Diagnostica mbH, Wiesbaden, Germany). The calculation of the optical density of the s Shanghai Shanghaiamples was carried out at a wavelength of 593 nm. Total cholesterol (CT) and triglycerides (TG) were analyzed using the enzymatic method (CHOD—PAP) (Biomagreb, Cholesterol, Ghod, PAP, Paris, France). The calculation of the optical density of the samples was carried out at a wavelength of 505 nm. Creatine kinase (CK) was analysed using the enzymatic assay method described by Olivier, and modified by Rosalski and then by Szasz by the monocarbonate CK-NAC IFCC reagent. The calculation of the optical density of the samples was carried out at a wavelength of 340 nm. The kinetic method was set at 340 nm for lactate dehydrogenase (LDH). For lactate ([La]), the enzymatic/colorimetric method (Roche COBAS, Roche, Basel, Switzerland) by the reagent LACT2-Lactate Gen.2 (Roche Diagnostics GmbH, Indianapolis, IN, USA) was used. The calculation of the optical density of the samples was carried out at a wavelength of 552 nm. Glucose (GLC) was analyzed using the enzymatic photometric method in the presence of the enzymes glucose oxidase and peroxidase (GOD—PAP) (Biomagreb, Paris, France). The calculation of the optical density of the samples was carried out at a wavelength of 505 nm. Hemoglobin A1c (HbA_1c_), was analyzed by the chromatographic assay method using the GX Assay Kit that was designed for use with the Tosoh Glycohemoglobin Automated Analyzer (HLC-723GX). For gamma glutamyl transferase (γGT), the kinetic colorimetric method (IFCC) was used. The calculation of the optical density of the samples was carried out at a wavelength of 405 nm. Total bilirubin (BIL-T) was analyzed using the diazo reaction with sulphanilic acid and sodium nitrate. The calculation of the optical density of the samples was carried out at a wavelength of 555 nm.

Cortisol concentrations were analyzed by ELFA (enzyme-linked fluorescent assay; VIDAS, BioMerieux, Paris, France). Glucagon levels were determined by radioimmunoassay techniques (RIA). At the time of the assay, the samples were thawed before being analysed. The results are expressed in μg/mL or pmol/mL. The lowest level of glucagon that could be detected by this test was 18.453 ± 2 μg/mL with a 100 μL sample.

The quantitative determination of insulin was done by the immuno-enzymatic test of chemiluminescence (The Insulin Kit (Human) CLIA). The concentration of insulin was determined at 450 nm with a microtiter plate reader within 15 min. The sensitivity of the test was found to be 2.0 IUU/mL.

### 2.7. Statistical Analysis

The normality of the distribution of variables was tested using the Shapiro−Wilk test. All statistical tests were processed using STATISTICA software (StatSoft, Statistica 10) and averaged ± standard error (M ± ES) in text, tables, and figures. The data were analyzed using the general linear model ANOVA.

Performance following the 6MWT and RPE were compared using a two-factor ANOVA with repeated measures (2 (conditions: without vs. with supplementation) × 2 (supplement: PLA vs. OFIJ)) with repeated measures. The Student’s *t* test was used to compare OFIJ and PLA conditions. Cardiovascular parameters (i.e., HR, SBP, and DBP), markers of muscle damage (LDH, CK, and La), lipid profile (HDL, TC, TG, and LDL), HbA1c and GLC), liver parameters (γGT and total bilirubin), and hormonal responses (cortisol, insulin, and glucagon) were analyzed using a three-factor ANOVA (2 (conditions) × 2 (supplement) × 2 (exercise)) with repeated measures. The Bonferroni post hoc test was performed whenever significant effects or significant interaction were seen. To evaluate the practical significance of the data, effect sizes were calculated as partial eta-square (η_p_^2^). The test−retest reliability is expressed by interclass correlation coefficients (ICC). The level of statistical significance was set at *p* < 0.05.

## 3. Results

### 3.1. The 6 Min Walking Test and RPE Scores

The ANOVA showed no significant effects for condition (F = 0.45, η_p_^2^ = 0.02, *p* > 0.05) and supplement (F = 1.36, η_p_^2^ = 0.70, *p* > 0.05). Similarly, no significant interaction conditions × supplement (F = 0.33, η_p_^2^ = 0.01, *p* > 0.05) was observed between these two factors. 

For the RPE scores (Figure 2), the ANOVA showed significant condition (F = 13.84, η_p_^2^ = 0.60, *p* < 0.01) and supplement (F = 17.01, η_p_^2^ = 0.65, *p* < 0.01) effects. Similarly, a significant interaction condition × supplement (F = 8.11, η_p_^2^ = 0.47, *p* < 0.05) was observed. The post hoc analysis showed that RPE scores recorded after the 6MWT were significantly lower following 4 days of supplementation for OFIJ (*p* < 0.01) compared to PLA (–2% vs. −14.39%, respectively).

### 3.2. Cardiovascular Parameters

The ANOVA showed significant exercise effect for HR (F = 179.17, η_p_^2^ = 0.90, *p* < 0.001), DBP (F = 626.11, η_p_^2^ = 0.98, *p* < 0.001), and SBP (F = 427.26, η_p_^2^ = 0.97, *p* < 0.001). However, no significant effect for condition or supplement was observed. Similarly, the interaction condition × supplement × exercise was not significant for HR, DBP, or SBP (F = 0.77, η_p_^2^ = 0.04, *p* > 0.05; F = 0.02, η_p_^2^ = 0.001, *p* > 0.05; F = 0.02 η_p_^2^ = 0.002, *p* > 0.05, respectively).

The post hoc analysis showed that HR, DBP, and SBP (Figure 3) recorded following the 6MWT were significantly higher than those recorded at rest in OFIJ and PLA without (+16.30% and +16.58% for HR, +16.70% and +16.34% for SBP, and +16.68% and +16.04% for DBP, respectively; *p* < 0.05) and with supplementation (+19.92% and +16.60% for CF, +14.15% and +14.07% for SBP, and +18.54% and +17.93% for DBP, respectively; *p* < 0.05).

### 3.3. Biochemical Markers

#### 3.3.1. Lipid Profile

Lipid profile results are showed in Figure 4. No significant condition (F = 0.49, η_p_^2^ = 0.04, *p* > 0.05; F = 0.07, η_p_^2^ = 0.008, *p* > 0.05; and F = 0.14, η_p_^2^ = 0.015, *p* > 0.05, for CT, LDL, and TG, respectively); supplement (F = 1.28, η_p_^2^ = 0.12, *p* < 0.05; F = 0.18, η_p_^2^ = 0.02, *p* > 0.05 and F = 0.21, η_p_^2^ = 0.02, *p* > 0.05, for CT, LDL, and TG, respectively); and exercise (F = 2.23, η_p_^2^ = 0.19, *p* < 0.01; F = 0.63, η_p_^2^ = 0.065, *p* > 0.05; F = 1.48, η_p_^2^ = 0.14, *p* > 0.05, for CT, LDL, and TG, respectively), effects were reported. Similarly, our analysis showed a no significant interaction condition × supplement × exercise (F = 1.62, η_p_^2^ = 0.15, *p* > 0.01; F = 0.29, η_p_^2^ = 0.03, *p* > 0.05; and F = 0.14, η_p_^2^ = 0.01, *p* > 0.05, for CT, LDL, and TG, respectively).

For HDL, ANOVA showed no significant condition (F = 0.43, η_p_^2^ = 0.01, *p* > 0.05) and supplement (F = 2.43, η_p_^2^ = 0.21, *p* > 0.05) effects. The analysis reported a significant exercise effect (F = 72.24, η_p_^2^ = 0.88, *p* < 0.001), but the interaction condition × supplement × exercise was not significant (F = 0.007, η_p_^2^ = 0.0007, *p* > 0.05).

Post hoc analysis revealed that HDL concentrations increased significantly after, compared to before, exercise (*p* < 0.01) during without and with supplementation for PLA and OFIJ (4.72% and 3.80%, respectively).

#### 3.3.2. Glycemia and HbA1c

For glycemia, ANOVA showed significant condition (F = 17.72, η_p_^2^ = 0.66, *p* < 0.01), supplement (F = 5.29, η_p_^2^ = 0.37, *p* < 0.01), and exercise (F = 6.66, η_p_^2^ = 0.42, *p* < 0.01) effects. On the other hand, no significant interaction condition × supplement × exercise (F = 0.09, η_p_^2^ = 0.009, *p* > 0.05) was observed.

For HbA1c, no significant condition (F = 1.55 η_p_^2^ = 0.14, *p* > 0.05), supplement (F = 3.54, η_p_^2^ = 0.28, *p* > 0.05), and exercise (F = 1.14, η_p_^2^ = 0.11, *p* > 0.05) effects were observed. Similarly, no significant interaction condition × supplement × exercise (F = 0.082, η_p_^2^ = 0.009, *p* > 0.05) was registered.

Post hoc analysis showed that glucose concentrations increased significantly after, compared to before, exercise (*p* < 0.01) for OFIJ (+2.01% before supplementation and +1.55% after supplementation) as well as PLA (+1.81% before supplementation and +0.99% after supplementation). Regarding the effect of supplementation on glycemia concentrations, a significant decrease was observed for OFIJ compared to PLA (*p* < 0.01) after the 6MWT during without and with supplementation (−1.92% vs. −3.11%, respectively).

#### 3.3.3. Hormonal Responses

For insulin and glucagon (Figure 5), the ANOVA showed no significant condition (F = 0.01, η_p_^2^ = 0.0006, *p* > 0.05 and F = 0.66, η_p_^2^ = 0.06, *p* > 0.05, respectively), and supplement (F = 1.07; η_p_^2^ = 0.10, *p* > 0.05 and F = 2.72, η_p_^2^ = 0.23, *p* > 0.05, respectively), effects. However, a significant exercise effect (F = 45.06, η_p_^2^ = 0.83, *p* < 0.001 for insulin, F = 42.65, η_p_^2^ = 0.82, *p* < 0.001 for glucagon) was observed. No significant interaction condition × supplement × exercise was observed (F = 0.04, η_p_^2^ = 0.0004, *p <* 0.05; F = 1.81, η_p_^2^ = 0.16, *p* > 0.05, for insulin and glucagon, respectively). The post hoc test revealed that insulin and glucagon concentrations increased significantly after, compared to before, exercise during both without supplementation (*p <* 0.01) (for OFIJ: +2.66% and +8.31%, respectively, and for PLA: +2.42% and +10.49%, respectively), and with supplementation (for OFIJ: +1.36% and +10.70%, respectively, and for PLA: +0.68% and +6.38%, respectively).

Regarding cortisol, no significant exercise effect (F = 0.2, η_p_^2^ = 0.02, *p* > 0.05) was observed. However, a significant condition (F = 44.3, η_p_^2^ = 0.83, *p <* 0.001) and supplement (F = 28.2, η_p_^2^ = 0.75, *p <* 0.001) effect was reported. No significant interaction condition × supplement × exercise (F = 0.1, η_p_^2^ = 0.009, *p* > 0.05) was registered. In addition, regarding the effect of supplementation, the ANOVA showed that cortisol levels increased significantly during supplementation with compared to without supplementation, before and after the 6MWT (0.89% vs. 0.32% at rest and 0.67% vs. 0.47% after 6MWT, for OFIJ, *p <* 0.01), while no significant changes were observed for PLA (*p* > 0.05).

#### 3.3.4. Muscle-Damage Markers

The results revealed that the levels of [La], CK, and LDH (Figure 6) increased significantly after the 6MWT (F = 202.37, η_p_^2^ = 0.95, *p <* 0.01 and F = 10.03, η_p_^2^ = 0.52, *p <* 0.01; F = 33.23, η_p_^2^ = 0.78, *p <* 0.01, respectively), in all test sessions. In addition, regarding the effect of supplement on these parameters, the ANOVA revealed a significant effect for CK and LDH levels (F = 11.15, η_p_^2^ = 0.55, *p <* 0.01 and F = 10.70, η_p_^2^ = 0.54, *p <* 0.01). Indeed, CK and LDH levels were lower during OFIJ compared to PLA (*p <* 0.05) during the with supplementation sessions. For [La] levels, no significant effect of supplement (F = 2.68, η_p_^2^ = 0.22, *p* > 0.05) was observed.

#### 3.3.5. Hepatic Parameters

For γGT and total bilirubin, the ANOVA showed a significant condition (Figure 7) (F = 5.37, η_p_^2^ = 0.37, *p <* 0.01 and F = 107.99, η_p_^2^ = 0.92, *p <* 0.001, respectively), supplement (F = 5.13 η_p_^2^ = 0.36 *p <* 0.01 and F = 191.19, η_p_^2^ = 0.95, *p <* 0.01, respectively), and exercise (F = 5.91, η_p_^2^ = 0.63, *p <* 0.01, F = 261.71, η_p_^2^ = 0.96, *p <* 0.01, respectively), effect. However, no significant interaction condition × supplement × exercise (F = 0.01, η_p_^2^ = 0.001, *p* > 0.05) was observed.

The post hoc analysis showed that γGT showed a significant increase after compared to before exercise (*p <* 0.01) and oxidative stress markers and total bilirubin levels showed a significant decrease after compared to before exercise (*p <* 0.01) in both OFIJ and PLA (for OFIJ, γGT, +7.20% and +9.13% without supplementation and with supplementation, respectively, for total bilirubin, −29.83% and −13.44% without supplementation and with supplementation, respectively, for PLA +5.52% without supplementation and +6.13% with supplementation for γGT and −24.76% without supplementation and −37.59% with supplementation for total bilirubin). Relating to the effect of supplementation on γGT concentrations, a significant decrease (*p <* 0.01) was observed for OFIJ compared to PLA following the 6MWT with 12.69% vs. −2.25%, respectively.

## 4. Discussion

The aim of this study was to investigate whether OFIJ supplementation has a beneficial effect on physical performance as well as biochemical and physiological responses following a 6MWT in type 2 diabetic patients. Our results indicated that 4 days of OFIJ supplementation is beneficial for performance (small increase in performance and reduced RPE) during the 6MWT. In addition, our data showed that, following the DPPH test, OFIJ had an antioxidant capacity. Moreover, OFIJ reduced muscle-damage markers (i.e., CK and LDH) and the glucose level.

Several studies have shown that OFIJ has a greater antioxidant capacity than other fruits [11,24] as the cactus pear contains compounds with anti-radical action (i.e., phenolics, flavonoids, and pigment compounds). These compounds have been found to possess a high potential for capturing free radicals. OFIJ contain substances that can inhibit the action of free radicals [9,10,25,26].

In the present study, OFIJ supplementation is beneficial for performance (a small increase in performance with lower RPE) during the 6MWT. It is difficult to compare our results with those of other studies because the protocols differ in terms of the supplements used, the exercise assessment, and the study population. Some studies on nitrate supplementation in beet juice and physical performance have shown some beneficial changes [27,28], but others have shown no effect [29].

Regarding cardiovascular parameters, our results showed that blood pressure and heart rate increased significantly after the 6MWT in all conditions. Adults with type 2 diabetes are also characterized by an increase in exercise-related blood pressure [30]. In fact, patients with type 2 diabetes are characterized by high systolic blood pressure at rest and arterial stiffness [31]. On the other hand, OFIJ supplementation does not affect cardiovascular parameters in patients. Our results are in line with those of Gilchrist et al. [32] who examined the impact of dietary nitrate supplementation (antioxidant) in people with type 2 diabetes and found no statistical difference in the 24 h blood pressure. Similarly, Shepherd et al. [20] revealed that there was no difference between beet-juice supplementation and PLA for blood pressure. in addition, Suksomboon et al. [33] showed that, in diabetic patients, supplementation with the antioxidant coenzyme Q10 for 12 weeks did not affect blood pressure. On the other hand, after receiving 500 mg/day of aqueous extracts of cinnamon, systolic pressure was reduced in obese and diabetic subjects after 12 weeks of treatment [34].

With respect to liver parameters, our results showed that total γGt and bilirubin levels were affected by exercise in all sessions. Our results are similar to those of Swift et al. [35] and Loprinzi & Abbott [36] who reported that physical activity was associated with bilirubin increase in insulin-resistant adults, but not in insulin-sensitive adults. However, our results differ from those of Tanaka et al. [37] who reported no relationship between physical activity and bilirubin. Increased physical activity may promote activity of hemooxygnase-1 (HO-1) [38] which is the enzyme responsible for the conversion of bilirubin to biliverin. Similarly, a reduction in bilirubin levels has been associated with insulin resistance states such as the metabolic syndrome [39] and type 2 diabetes [40].

In addition, plasma γGT is considered as a marker of oxidative stress. However, it is well-accepted that physical exercise leads to alterations in physiological responses, especially for markers of oxidative stress [11], which is accentuated in diabetic patients and is also implicated in the pathogenesis of diabetic complications [41]. Indeed, oxidative stress increases the plasma activity of γGT [42], which coincides with our results. With respect to supplementation, our data revealed that OFIJ supplementation reduced bilirubin and γGT concentrations before and after the 6MWT. Consistent with our study, in women with type 2 diabetes, vitamin E and C supplementation for 6 weeks resulted in a significant reduction in bilirubin levels [43,44]. As indicated by other studies, OFIJ is rich in flavonoids and vitamins which in turn serve as major species of antioxidant activity capable of trapping free radicals released in excess during aerobic exercise [45]. Our results have confirmed previous studies in the field that have investigated the protective effect of fruit polyphenols against exercise-induced oxidative stress [46,47]. In this sense, Banini et al. [48] suggested that antioxidant supplementation also reduces exercise-induced oxidative stress, particularly with a moderate exercise regimen.

Our data revealed that among biochemical measurements, only GLC and HDL were significantly increased after a 6MWT for both PLA and OFIJ, while HbA1c, CT, TG, and LDL did not vary with exercise during without or with supplementation sessions. Regular exercise changes lipid fractions in a favorable direction: decreased triglycerides and elevated HDL-cholesterol [49]. A previous study [50] has reported low-density lipoprotein cholesterol reductions after aerobic and resistance training. Another study showed a tendency to reduce triglyceride levels after resistance training alone [51], while others found no change after aerobic, resistance, or combined training [52]. A previous study has shown a decrease in blood glucose [53] after exercise. In accordance with the present results, Ribeiro et al. [54] reported no change in HbA1c while Andrade-Rodríguez et al. [55] reported a decrease in HbA1c values. Exercise intensity, volume, and frequency are associated with reductions in HbA1c [56]. In fact, Ribeiro et al. [54] reported that groups exercising for 4 months (diabetic and nondiabetic) showed no change in plasma levels of total cholesterol, LDL-cholesterol (LDL), HDL-cholesterol (HDL), triglyceride (TG), GLC, or insulin compared to sedentary diabetic groups. With respect to supplementation, our results suggested that biochemical measures (i.e., lipid profile (LDL, HDL, TG, CT, and HbA1c) are not affected by OFIJ supplementation at baseline (except the GLC rate which was reduced following supplementation). In this context, Giglio et al. [57] reported that 4 weeks of OFIJ supplementation improved metabolic parameters and reduced atherogenic small dense LDL in patients with risk factors for the metabolic syndrome. Our results agree with previous studies. Indeed, López-Romero et al. [58] reported that OFIJ consumption can reduce postprandial glucose and serum insulin and increase antioxidant activity in both healthy and diabetic people. This reduction in blood glucose was more explicit after exercise than at rest, which coincides with our data. In addition, the inclusion of OFIJ in the diet in patients with type 2 diabetes reduced the blood glucose by 36 mg/dL (2 mmol/L), reaching a level close to that recommended by the European Diabetes Policy Group [59].

In accordance with our results, Fukino et al. [60] showed that a green-tea extract administered for 2 months also had no effect on HbA1c level in diabetic subjects. In addition, resveratrol, a polyphenol present in fruits, did not induce an improvement in lipid variables but slightly increased the circulating concentrations of total cholesterol and triglycerides at a dose of 500 mg [61]. However, the study of Mang et al. [62] revealed that 3 g/day of powder for 4 months decreased glycosylated haemoglobin and improved the lipid profile in type 2 diabetes. Concerning the stability of HbA1c levels and lipid profile concentrations with OFIJ supplementation, this effect can be explained by the low dose of fresh juice that seemed insufficient to exert a direct effect on these parameters.

With respect to hormonal responses, except for cortisol, our data showed that insulin and glucagon concentrations were significantly increased after the 6MWT for PLA and OFIJ. Our data are in line with those of Sigal et al. [63] who showed that exercise increases insulin sensitivity by stimulating glycogenolysis and glucose uptake by the muscle, thus reducing insulin levels in the blood. In addition, under the condition of 90% of VO2max in young individuals, it has been shown that blood hormone levels increased [64]. Regarding the hormonal parameters (i.e., insulin and glucagon), the authors showed no significant alteration with either PLA supplementation or with OFIJ [64]. Regarding cortisol, it was affected by supplementation before and after the 6MWT, but this alteration was significant only for OFIJ. Christou et al. [65] suggested that short-term supplementation of OFIJ extract in subjects with normal glucose tolerance, prediabetes, or diabetes did not affect the rates of insulin during the oral glucose-tolerance test. One study showed that 1000 mg of OFIJ extract +3 g leucine increased plasma insulin concentrations after a training program and thus potentially accelerated glycogen resynthesis. In addition, according to López-Romero et al. [58], consumption of OFIJ could reduce plasma insulin in type 2 diabetics. Regarding cortisol, Cinar et al. [66] reported that magnesium supplementation of 10 mg per kilogram of body weight for one month, if combined with physical activity, could result in increased cortisol levels in individuals.

Our results revealed that levels of CK, LDH and [La] increased significantly after the 6MWT. For LDH and LDL, Giglio et al. [57] reported that OFIJ had a beneficial effect on LDL and HDL. These results confirm a previous study that showed a significant elevation of plasma CK levels after exercise [67]. Similarly, with moderate-intensity exercise, lactate levels may increase slightly, as we have observed, and the activity of the LDH increases and may offset the production of hydrogen ions to delay muscle fatigue [68]. OFIJ supplementation leads to a significant decrease in muscle damage accompanied by a reduction in RPE scores before and after the 6MWT. It is difficult to compare our results with those of the literature as we were the first to evaluate the effect of OFIJ supplementation on physiological parameters in diabetic patients following exercise. Regarding CK levels, no work on diabetic patients has been undertaken, although work has been done on animals. This showed that antioxidant supplementation reduced CK levels [69].

Despite the importance of the results of the present study, some limits should be highlighted in this study. Concerning the oxidative system, we assayed only γGT. The use of other markers for the evaluation of oxidative stress (i.e., MDA and F2-isoprostane) could be of interest as we can better control the oxidant–antioxidant balance after supplementation with OFIJ. In addition, the small sample size could be a limitation for the study. Furthermore, in the future studies, a control group (not suffering from type 2 diabetes) could be helpful for the comparison of the between groups responses.

## 5. Conclusions

Following the DPPH test we determined that OFIJ has an antioxidant capacity. In addition, our data showed that 4 days of OFIJ supplementation was beneficial for performance (there was a small increase in performance and reduced RPE) during the 6MWT. Moreover, OFIJ reduced muscle-damage markers (i.e., CK and LDH) and glucose levels.

## Figures and Tables

**Figure 1 medicina-58-01561-f001:**
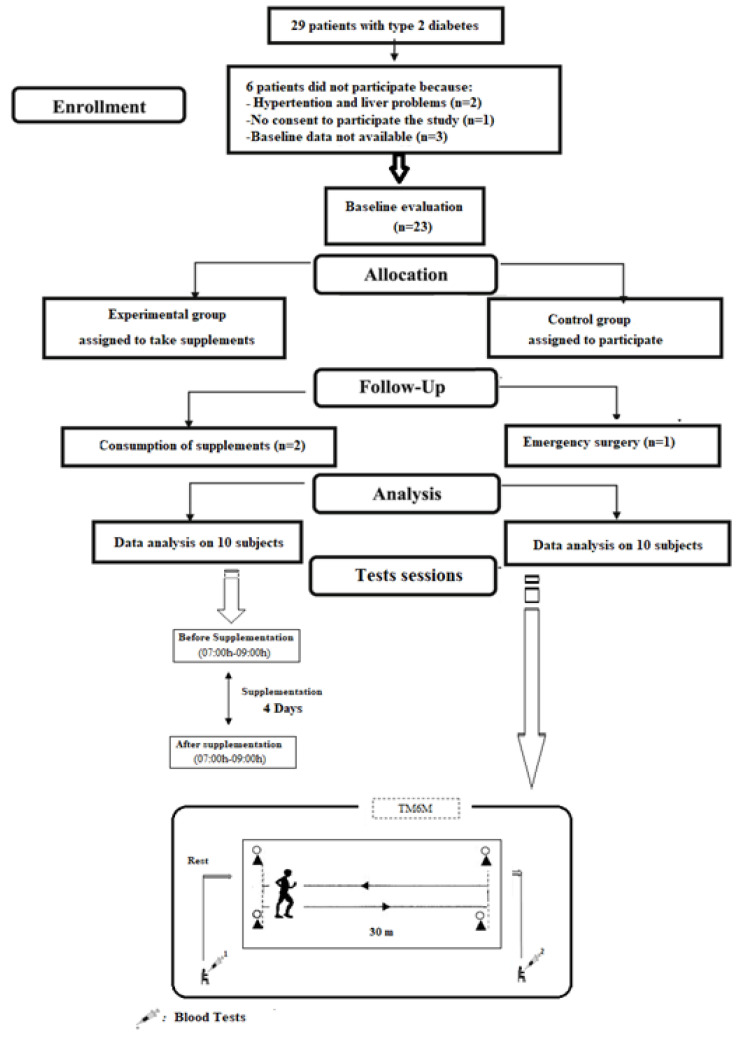
Flow-chart of participant recruitment.

**Figure 2 medicina-58-01561-f002:**
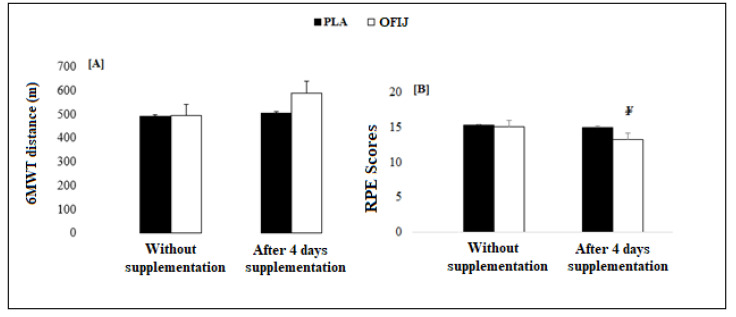
6MWT [**A**] and RPE scores [**B**] (mean ± SD) without and with supplementation of PLA and OFIJ. ¥: significant difference compared with PLA (*p* < 0.05).

**Figure 3 medicina-58-01561-f003:**
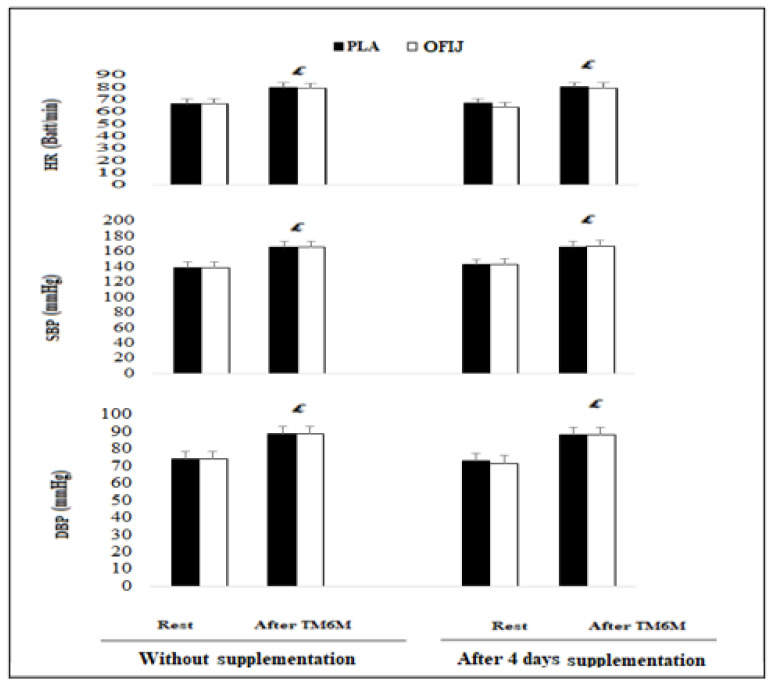
Cardiovascular parameters (mean ± SD; n = 20) recorded before and after the 6MWT without and with supplementation of PLA and OFIJ. £: significant difference compared to before the 6MWT (*p* < 0.05).

**Figure 4 medicina-58-01561-f004:**
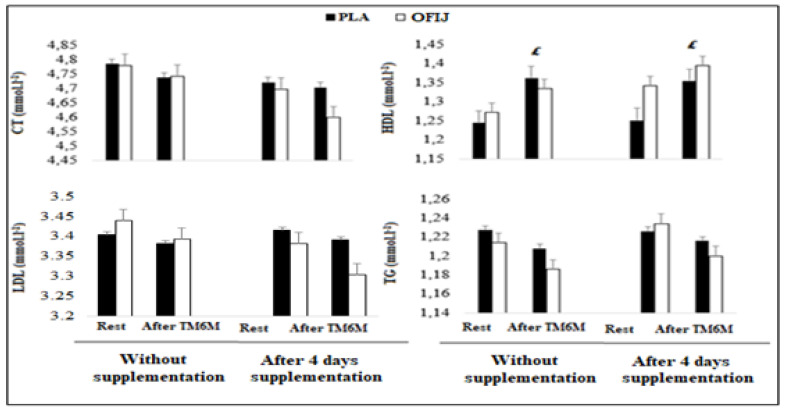
Lipid-profile parameters (mean ± SD; n = 20) recorded before and after the 6MWT during without and with supplementation of PLA and OFIJ. £: significant difference compared to before the 6MWT (*p* < 0.05).

**Figure 5 medicina-58-01561-f005:**
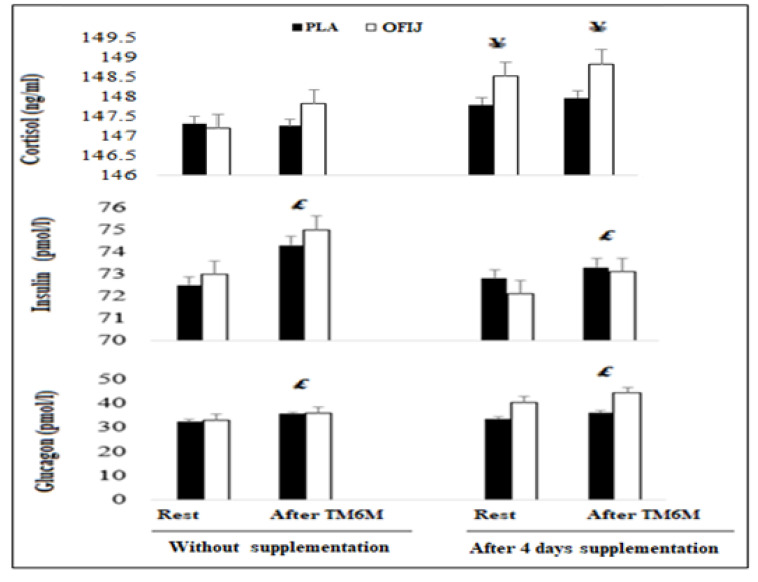
Hormonal parameters (mean ± SD; n = 20) recorded before and after the 6MWT during without and with supplementation of PLA and OFIJ. £: significant difference compared to before the 6MWT (*p <* 0.05). ¥: significant difference compared to without supplementation (*p* < 0.05).

**Figure 6 medicina-58-01561-f006:**
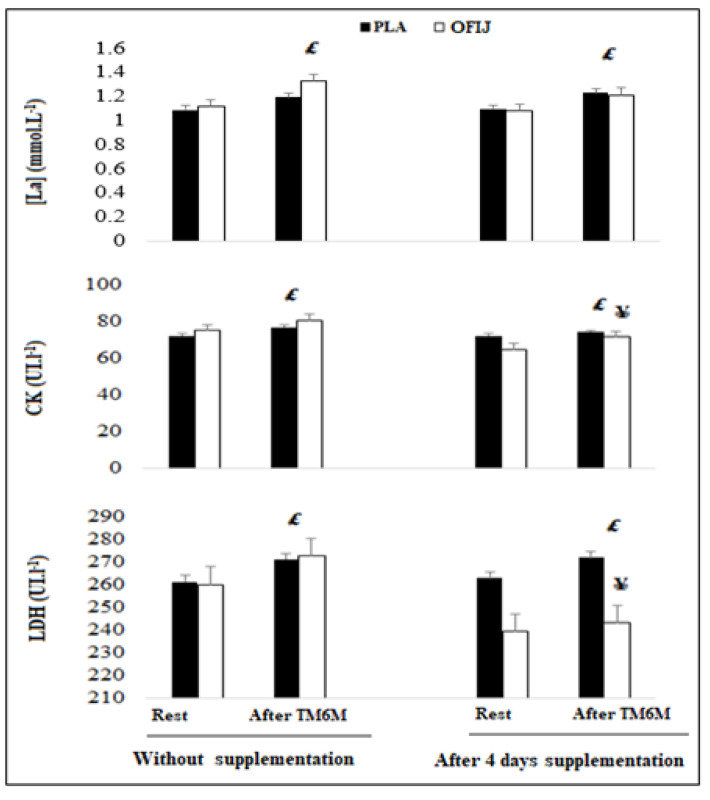
Muscle-damage markers (mean ± SD; n = 20) recorded before and after the 6MWT during without and with supplementation of PLA and OFIJ. £: significant difference compared to before the 6MWT (*p <* 0.05). ¥: significant difference compared to without supplementation (*p <* 0.05).

**Figure 7 medicina-58-01561-f007:**
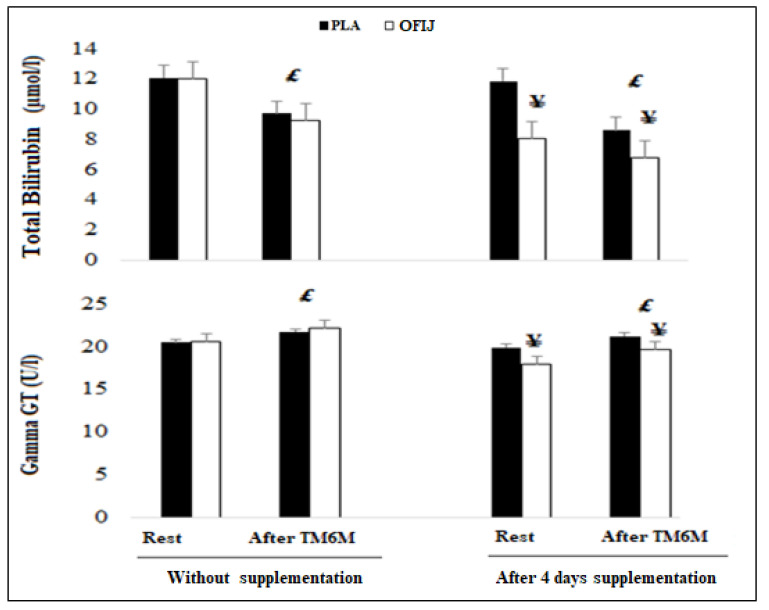
Hepatic parameters (mean ± SD; n = 20) recorded before and after the 6MWT during without and with supplementation of PLA and OFIJ. £: significant difference compared to before the 6MWT (*p <* 0.05). ¥: significant difference compared to without supplementation (*p <* 0.05).

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
