# Peer review of "Does Opuntia ficus-indica Juice Supplementation Improve Biochemical and Cardiovascular Response to a 6-Minute Walk Test in Type 2 Diabetic Patients?"

_medicina, 2022, doi:10.3390/medicina58111561_

Round 1

Reviewer 1 Report

The article's subject is interesting. The topic is original, in the literature there are several articles on the antioxidant property of Opuntia ficus-indica juice and its relationship with oxidative stress and diabetes. The paper is well written, the text is clear and i read easy and understood well.Evidence,discussion,conclusions are consistent with the study purpose and results, the subject is well compared. The article was written and discussed on the main topic asked.

Author Response

We thank the reviewer for his/her thorough review of our work.

Reviewer 2 Report

In the exclusion criteria could be include whether patients take hypoglycemic drugs

The n of patients is very small

In all the text change de liter abbreviation (l to L) ml to mL etc

Line 132 change NM to nm

The conclusion should be restructured, since mentions that inflammatory parameters could be measured, but it is not clear

Author Response

In the exclusion criteria could be include whether patients take hypoglycemic drugs

This exclusion criteria was added. Please see changes made in the revised version.

 The n of patients is very small

The reviewer is right. We have added this as a limitation on the revised manuscript.

 In all the text change de liter abbreviation (l to L) ml to mL etc

Change made in the new version. Please see the new text.

 Line 132 change NM to nm

Change made in the new version. Please see the new text.

The conclusion should be restructured, since mentions that inflammatory parameters could be measured, but it is not clear

Sorry for the lack of precision. Correction made in the new version.

Reviewer 3 Report

11.      Authors declared that  the aim of their study was to evaluate the effect of Opuntia ficus-indica juice (OFIJ) on performance and physiological responses to a 6-minutes walking test (6MWT) patients with type 2 diabetes. In the introduction they exposed the problem of diabetes over the world and underlined the role of sport activity in the improvement of glycaemic control and reduction of diabetes consequences. In the other hand they emphasized - but not very fortunately in my opinion - that physical exercise also increase the production of reactive oxygen species (ROS) by cells, which can lead to cellular damage, and in the next sentence they wrote that these ROS are considered in the literature as an indicator of powerful oxidative stress in biological systems. This is, in my opinion, a very dangerous and false simplification that leads to the conclusion that physical activity is harmful. Moreover, diabetes itself induces free radicals.

2.      Authors also described beneficial effects of OFIJ. They hypothesized that the OFIJ supplementation is a potent strategy to improve biochemical and physiological responses to a 6-minute walk test in type 2 diabetic patients. But the Authors' hypothesis is surprising to me – they hypothesized that the OFIJ supplementation is a potent strategy to improve biochemical and physiological responses to a 6-minute walk test in type 2 diabetic patients. Does this mean that, according to the Authors, 6MWT is a disadvantageous effect for these patients and that OFIJ would compensate for these effects?

3.      The Authors wrote that the 6MWT is a sub-maximal exercise test used to assess aerobic performance. They did not explain why they chose such a test for their experiment, since it is not intended for this type of research.

The Authors did not plan a control group that was not suffering from Type 2 diabetes. It was a huge mistake – the Authors in the discussion referred to the results of other researchers who included such a control group. Moreover in the discussion the following sentence line 459-462 was written: ”Moreover, it has promoted the reduction of muscle damage markers (i.e. CK and LDH) and the glucose level, and it minimize the oxidative stress (i.e. the rate of lipid peroxidation: γGT) induced by exercise and by the pathology of diabetes.”  Authors cannot draw a conclusion that OFIJ supplementation reduced effects of the pathology of diabetes, because they did not have an appropriate non-diabetic control group. Without health control group all results are not reliable.

4.      The initial glycaemia of the patients studied in the experiment has not been described.

5.      What were the exact ingredients of placebo?

6.      On what basis were the blood parameters selected for the tests?

The parameters related to the work of the cardiovascular system, selected by the Authors, are known to increase after exercise, and it is obvious. If the extract was expected to change them in such a short time, it would mean that it does not have health-promoting properties. Moreover, hemoglobin HbA1c does not change over such a short period of time. Moreover, could parameters such as CT, LDL and HDL change in such a short time - if HDL changed why CT and not LDL - why CT did not change?

LA increase after activity is also obvious and is not connected with muscle damage, what was suggested by the Authors.

7.      References to the results of other researchers in the discussion are not adequate and do not explain anything. The conditions of the experiments conducted by other researchers were very different - it cannot be written that the results are similar. In addition, the authors wrote, for example, that their cost of oxygen results are similar to others, and that this indicator has not been studied at all.

8.      What was the significance of the increase in glucose after 6MWT and the influence of OFIJ on this parameter? The increase in glucose may have been caused by the effect of short exercise time on the increase in adrenaline concentration. On the other hand, insulin concentration after activity was increased, - thus – why glucose was also higher? Moreover, longer physical activity eliminates glucose from the blood and affects, as the authors emphasized in the introduction, on the improvement of glycaemic control. What is the explanation of these results?

9.      Did the Authors take into account the fact that they conducted the experiment for 4 days? If the results refer to experiments with 4, this should be marked on the graph. The description of the used statistic methods also does not indicate that 4 days of experience were taken into account.

10.   Conclusions appear to be untrue. The authors concluded that OFIJ supplementation following the 6MWT in diabetic patients is a therapeutic strategy for diabetic subjects that contribute to reduce glucose levels, markers of muscle damage and exercise-induced oxidative stress. In summary, according to Authors, OFIJ reduced the harmful effects of 6MWT in patients with type 2 diabetes. It is improper conclusion.

11.   Abbreviations used for the first time should be explained, Authors should pay attention to the notation of units, English grammar.

Moreover in lines:

172 – “The calculation of the optical density of the samples is carried out at a wavelength  of 593 nm.”  – the past tense should be used

174 – “The calculation of the optical density of the samples is carried out at a wavelength of 505 nm” – the past tense should be used

Author Response

Point-by-point response to the reviewers

 We thank the reviewers and the editor for their thorough review of our work and for the very constructive and helpful comments. We have considered the comments and have provided specific responses for each remark. Our responses appear in red typeface. We hope that this version has been improved and that is now suitable for publication in your journal.

Reviewer 3

  1. Authors declared that  the aim of their study was to evaluate the effect of Opuntia ficus-indica juice (OFIJ) on performance and physiological responses to a 6-minutes walking test (6MWT) patients with type 2 diabetes. In the introduction they exposed the problem of diabetes over the world and underlined the role of sport activity in the improvement of glycaemic control and reduction of diabetes consequences. In the other hand they emphasized - but not very fortunately in my opinion - that physical exercise also increase the production of reactive oxygen species (ROS) by cells, which can lead to cellular damage, and in the next sentence they wrote that these ROS are considered in the literature as an indicator of powerful oxidative stress in biological systems. This is, in my opinion, a very dangerous and false simplification that leads to the conclusion that physical activity is harmful. Moreover, diabetes itself induces free radicals.

The reviewer is right. This part was reworded as follow:

Also, moderate physical activity combined with a better dietary balance can delay the onset of type 2 diabetes [4]. However, it is well established that physical exercise increase the production of reactive oxygen species (ROS) by cells, which can lead to cellular damage [5,6,7].

Please see changes made in the introduction section.

  1. Authors also described beneficial effects of OFIJ. They hypothesized that the OFIJ supplementation is a potent strategy to improve biochemical and physiological responses to a 6-minute walk test in type 2 diabetic patients. But the Authors' hypothesis is surprising to me – they hypothesized that the OFIJ supplementation is a potent strategy to improve biochemical and physiological responses to a 6-minute walk test in type 2 diabetic patients. Does this mean that, according to the Authors, 6MWT is a disadvantageous effect for these patients and that OFIJ would compensate for these effects?

The hypothesis means that the OFIJ could help the patient to perform better in the 6MWT (perform better distance).

The hypothesis was reformulated as follow:

We hypothesize that the OFIJ supplementation is beneficial to perform greater distance during the 6-minute walk test and to observer better biochemical and physiological responses in type 2 diabetic patients.

I hope that this is clearer now.

  1. The Authors wrote that the 6MWT is a sub-maximal exercise test used to assess aerobic performance. They did not explain why they chose such a test for their experiment, since it is not intended for this type of research.

The Authors did not plan a control group that was not suffering from Type 2 diabetes. It was a huge mistake – the Authors in the discussion referred to the results of other researchers who included such a control group. Moreover in the discussion the following sentence line 459-462 was written: ”Moreover, it has promoted the reduction of muscle damage markers (i.e. CK and LDH) and the glucose level, and it minimize the oxidative stress (i.e. the rate of lipid peroxidation: γGT) induced by exercise and by the pathology of diabetes.”  Authors cannot draw a conclusion that OFIJ supplementation reduced effects of the pathology of diabetes, because they did not have an appropriate non-diabetic control group. Without health control group all results are not reliable.

The following explanation for the 6MWT was added: “The 6MWT was selected in the present study as it is a submaximal aerobic exercise that could be performed by patients with type 2 diabetes.”

For the second point, the reviwer is right we adjusted our conclusions. Our main objective was to examine the effect of OFIJ on the patients. This point was added the absence of a control group as a limitation for the study.

The following sentence was added:

Furthermore, in the future studies, a control group (not suffering from Type 2 diabetes) could be helpful for the comparison of the between groups responses.”

  1. The initial glycaemia of the patients studied in the experiment has not been described.

This information was added. Please see changes made in the text.

  1. What were the exact ingredients of placebo?

The placebo was a fruit concentrate of isoenergetic origin synthetically designed to have a similar consistency and color, but without the phytochemical content of the OFIJ.

  1. On what basis were the blood parameters selected for the tests?

The parameters related to the work of the cardiovascular system, selected by the Authors, are known to increase after exercise, and it is obvious. If the extract was expected to change them in such a short time, it would mean that it does not have health-promoting properties. Moreover, hemoglobin HbA1c does not change over such a short period of time. Moreover, could parameters such as CT, LDL and HDL change in such a short time - if HDL changed why CT and not LDL - why CT did not change?

LA increase after activity is also obvious and is not connected with muscle damage, what was suggested by the Authors.

The reviewer is right. However, our data reported all these changes. We have changed and removed some parts of the discussion.

Please see changes made in the text.

  1. References to the results of other researchers in the discussion are not adequate and do not explain anything. The conditions of the experiments conducted by other researchers were very different - it cannot be written that the results are similar. In addition, the authors wrote, for example, that their cost of oxygen results are similar to others, and that this indicator has not been studied at all.

A substantial revision of the discussion part was realized. Please see changes made in the revised version.

  1. What was the significance of the increase in glucose after 6MWT and the influence of OFIJ on this parameter? The increase in glucose may have been caused by the effect of short exercise time on the increase in adrenaline concentration. On the other hand, insulin concentration after activity was increased, - thus – why glucose was also higher? Moreover, longer physical activity eliminates glucose from the blood and affects, as the authors emphasized in the introduction, on the improvement of glycaemic control. What is the explanation of these results?

The reviewer is right. The increase of glucose could be related to the increase of the adrenaline concentration (unfortunately not measured in the study).

  1. Did the Authors take into account the fact that they conducted the experiment for 4 days? If the results refer to experiments with 4, this should be marked on the graph. The description of the used statistic methods also does not indicate that 4 days of experience were taken into account.

Corrected in all graph. Please see the new version

 Conclusions appear to be untrue. The authors concluded that OFIJ supplementation following the 6MWT in diabetic patients is a therapeutic strategy for diabetic subjects that contribute to reduce glucose levels, markers of muscle damage and exercise-induced oxidative stress. In summary, according to Authors, OFIJ reduced the harmful effects of 6MWT in patients with type 2 diabetes. It is improper conclusion.

The conclusion was changed as follow:

Following the DPPH test, the OFIJ has an antioxidant capacity for catching free radicals. In addition, 4 days OFIJ supplementation improved performance of the 6MWT. Moreover, OFIJ reduce muscle damage markers (i.e. CK and LDH) and the glucose level. These results suggest that OFIJ supplementation is helpful for a submaximal physical exercise and for the some biochemical and physiological responses to the 6MWT.

Please see changes made in the revised version.

 Abbreviations used for the first time should be explained, Authors should pay attention to the notation of units, English grammar.

Corrected. Please see the new version.

 Moreover in lines:

172 – “The calculation of the optical density of the samples is carried out at a wavelength  of 593 nm.”  – the past tense should be used

Corrected. Please see the new version.

 174 – “The calculation of the optical density of the samples is carried out at a wavelength of 505 nm” – the past tense should be used

Corrected. Please see the new version.

Round 2

Reviewer 2 Report

Accept 

Author Response

(The authors gave the same response as above.)

Reviewer 3 Report

After reviewing the Authors' responses and the changes introduced, I still maintain my opinion and believe that the manuscript should be rejected. The improvement is insufficient (hypothesis and especially the disussion). In the context of the lack of a group of healthy people as a control, the research hypothesis cannot be confirmed. Moreover, the research hypothesis is still not convincing - the Authors wrote that it assumes that OFIJ is beneficial to observe (not observed) better biochemical and physiological responses in type 2 diabetic patients, but it is not known that parameters after 6MWT are worse, because there is not the control group.

Author Response

Authors are thankful to the editor and reviewer for their insightful review of the manuscript and for the very constructive and helpful comments. We have reconsidered the comments and have provided specific responses for each remark.

Reviewer 3 report

After reviewing the Authors' responses and the changes introduced, I still maintain my opinion and believe that the manuscript should be rejected. The improvement is insufficient (hypothesis and especially the disussion). In the context of the lack of a group of healthy people as a control, the research hypothesis cannot be confirmed. Moreover, the research hypothesis is still not convincing - the Authors wrote that it assumes that OFIJ is beneficial to observe (not observed) better biochemical and physiological responses in type 2 diabetic patients, but it is not known that parameters after 6MWT are worse, because there is not the control group.

Although the reviewer is right and the inclusion of a control group could be of importance in the present study (as presented in the limitation part), we suggest that our hypothesis could be verified when we compare one group with a PLA or OFIJ as our main objective was not the between groups comparison. The main aim was the between conditions comparison.

We hope that our responses and modifications of the manuscript are clearer and that the paper could be suitable for publication.
